**PLOS** NEGLECTED TROPICAL DISEASES

# Micro-CT visualization of a promastigote secretory gel (PSG) and parasite plug in the digestive tract of the sand fly *Lutzomyia longipalpis* infected with *Leishmania mexicana*

**Martin J. R. Hall**[1]*, **Debashis Ghosh**[2], **Daniel Martín-Vega**[1,3], **Brett Clark**[1], **Innes Clatworthy**[1], **Robert A. Cheke**[4], **Matthew E. Rogers**[5]

**1** Natural History Museum, London, United Kingdom, **2** International Centre for Diarrhoeal Disease Research, Bangladesh (icddr,b), Dhaka, Bangladesh, **3** Universidad de Alcalá, Alcalá de Henares (Madrid), Spain, **4** Natural Resources Institute, University of Greenwich, Chatham Maritime, United Kingdom, **5** London School of Hygiene and Tropical Medicine, London, United Kingdom

* m.hall@nhm.ac.uk

**Data Availability Statement:** All quantitative data on which this study are based are provided in the Tables within the manuscript. Additional qualitative

## Abstract

Leishmaniasis is a debilitating disease of the tropics, subtropics and southern Europe caused by *Leishmania* parasites that are transmitted during blood feeding by phlebotomine sand flies (Diptera: Psychodidae). Using non-invasive micro-computed tomography, we were able to visualize the impact of the laboratory model infection of *Lutzomyia longipalpis* with *Leishmania mexicana* and its response to a second blood meal. For the first time we were able to show in 3D the plug of promastigote secretory gel (PSG) and parasites in the distended midgut of whole infected sand flies and measure its volume in relation to that of the midgut. We were also able to measure the degree of opening of the stomodeal valve and demonstrate the extension of the PSG and parasites into the pharynx. Although our pilot study could only examine a few flies, it supports the hypothesis that a second, non-infected, blood meal enhances parasite transmission as we showed that the thoracic PSG-parasite plug in infected flies after a second blood meal was, on average, more than twice the volume of the plug in infected flies that did not have a second blood meal.

## Author summary

Leishmaniasis is a debilitating disease of the tropics, subtropics and southern Europe caused by protozoa of the genus *Leishmania*, that are transmitted by blood feeding sand flies. The parasites multiply and undergo morphological changes within the fly and have an effect on the fly that enhances their chances of transmission. One of these effects is the production of a glycoprotein-rich gel which, together with the parasites, forms a plug that blocks the sand fly gut and causes problems with feeding that enable more parasites to be transferred during biting. Using micro-computed tomography, for the first time we have been able to non-invasively visualize the plug in the fly in 3D and the way it distends the midgut and opens the stomodeal valve, facilitating transmission by regurgitation of the

data (sagittal section image files of each sand fly at the section showing maximum opening of the stomodeal valve) are available on the EMBL-EBI BioStudies database (Accession number S-BSST684, https://www.ebi.ac.uk/biostudies/studies/S-BSST684?query=S–BSST684). The full sagittal image stacks of each fly are also available upon request to the Micro-CT unit of the Natural History Museum by completion of the form at the following link, entering 'Feedback and General Enquiries' as the 'Subject': https://www.nhm.ac.uk/about-us/contact-enquiries/forms/emailform.jsp.

**Funding:** We are grateful to the Biotechnology and Biological Sciences Research Council, UK (https://bbsrc.ukri.org), for funding this project through the Gnatwork (grant reference number: BB/R005362/1 awarded to MJRH)(https://www.gnatwork.ac.uk). The funders had no role in study design, data collection and analysis, decision to publish, or preparation of the manuscript.

**Competing interests:** The authors have declared that no competing interests exist.

gut contents. We also show how a second, uninfected blood meal causes further extension of the midgut and, especially, the plug, supporting the hypothesis that such second meals can facilitate parasite transmission.

## Introduction

Phlebotomine sand flies (Diptera: Psychodidae) are vectors of the *Leishmania* (Trypanosomatida: Trypanosomatidae) parasites of vertebrates, including humans. Leishmaniasis is a neglected tropical disease (NTD) associated with poverty, causing a significant health, welfare and economic burden as well as delays to a wide range of development programmes in the tropics, with some 1.5–2.0 million new cases annually [1]. However, it is not only the vertebrate hosts that suffer as a consequence of infection as *Leishmania* can have a significant impact on the sand fly vector [2], with parasite production of a proteophosphoglycan-rich promastigote secretory gel (PSG) that can block the lumen of the anterior midgut [3]. Furthermore, it has been shown that second, uninfected blood meals, subsequent to the primary infectious blood meal, promote *Leishmania* replication, amplifying the numbers of parasites by up to 125-fold, thereby increasing sand fly infectivity [4,5]. Development of the parasite in the sand fly vector is complex, involving several different morphological forms. Amastigotes are ingested with an infected blood meal and these develop successively into a variety of flagellated promastigote forms: procyclics, nectomonads, leptomonads, haptomonads and the non-dividing, infective metacyclics [6,7]. Metacyclic promastigotes are produced by differentiation of leptomonad promastigotes in the anterior midgut, and the latter form also produces the PSG. It is the accumulation of PSG and a mass of parasites that obstructs the sand fly gut [8,9] (herein described as the 'PSG-parasite plug'), making it difficult for the fly to feed and, thereby, leading to more and longer feeding attempts, which increase the chances of parasite transmission–termed the "blocked fly hypothesis" [8,10].

In research of the infection of vectors with disease agents that cause NTDs, there is tremendous scope for visualization techniques to improve the fundamental understanding of parasite-vector interactions, the basis of vectorial competence and transmission. Precise spatiotemporal information on the progress of an infection can tell us much about the interactions required for successful colonization and transmission. Previously, PSG plugs have been removed from dissected midguts [9], but it has been difficult to visualize them other than through such invasive procedures. In this study we attempted to visualize the PSG inside intact sand flies by using micro-computed tomography (micro-CT). Micro-CT is a technique that is essentially non-invasive, requiring only fixation and, for study of internal soft tissues, staining of the specimen tissues before scanning of the entire insect. It is being used increasingly to study the internal morphology of a range of insects, including flies [11,12]. The technique enables volumetric measurement of discrete organs ranging in size from the entire digestive tract [13,14] to discrete neuropils of the brain [15]. Many of those studies have been of larger Diptera: to achieve sufficient resolution in the scanning of small Diptera is more challenging due to staining issues and the increased scan times needed. However, a recent study of the infection of *Simulium* blackflies with *Onchocerca* nematodes demonstrated the feasibility of using micro-CT to scan small flies, revealing a parasite in the flight muscles of the fly [16].

We studied *Leishmania* infections in a laboratory model, *Leishmania mexicana* infection of *Lutzomyia longipalpis*, using micro-CT to visualize and measure, for the first time by three dimensional (3D) reconstructions, the effect of different post-infection periods and of a subsequent blood meal on the sand fly midgut.

## Methods

### Ethics statement

All animal experiments were carried out in accordance with the UK's Animals (Scientific Procedures) Act 1986 (ASPA), which transposes European Directive 2010/63/EU into UK national law. The animal studies were approved by the UK Home Office in granting Project Licence number 70/8427 under the ASPA and all protocols had undergone appropriate local ethical review procedures by the Animal Welfare and Ethical Review Board (AWERB) of The London School of Hygiene and Tropical Medicine.

### Parasites

*Leishmania mexicana* (MNYC/BZ/62/M379) was cultured as previously described [3]. Cultures were initiated with 5 x $10^6$/ml promastigotes. For sand fly infection, axenic amastigotes were obtained by adding early-passage, mid-logarithmic phase promastigotes into acidified media [17]. To do this, promastigotes were grown in M199 medium (Invitrogen, supplemented with 1% penicillin-streptomycin (v/v), 1 x Basal Medium Eagle vitamins (v/v) and 10% heat-inactivated foetal calf serum (v/v)), pH 7.2 at 26˚C and passaged at 5x$10^5$/ml in acidified M199 medium (pH 5.5, supplemented as above) and grown for 4–5 days in a humidified incubator at 34˚C, 80% relative humidity (RH) and 5% $CO_2$.

### Sand fly rearing and infection

Day 3 post-emergent female *Lu. longipalpis* sand flies (Jacobina strain) were infected with *L. mexicana* axenic amastigotes by feeding through a chick skin membrane on heat-inactivated (h.i.) human blood (h.i. sera, inactivated at 56˚C for 45 min, combined with RBCs washed three times in PBS) that contained amastigotes at 2x$10^6$/ml. Blood-fed flies were separated into 20x20x20 cm netting cages and given *ad libitum* access to 10% (w/v) sucrose-soaked cotton wool placed on the top of the cage. Flies were maintained at 26–27˚C and 80% RH. On the 6th day of infection, flies were allowed to feed directly on an anaesthetized BALB/c mouse. Those sand flies that fed were separated and maintained for a further 6 days (12 days post-infection in total). Infected flies that did not feed a second time were also maintained, as the 'primary' infection.

For each group a random sample of 10–15 flies was sampled to assess infection by dissecting and homogenizing each midgut in 30 μl PBS. Ten microliters of the homogenate were fixed on a slide with methanol and stained with 10% Giemsa stain for 15 min at room temperature to check for infection. For specimen collection, flies were knocked down on ice and briefly washed in a weak detergent solution to remove hairs, then stored in 70% ethanol (v/v) at 4˚C until they were scanned.

The initial infective blood meal (**INF**) was considered to be at time **T0** and, for some, a second non-infected blood meal (denoted by +) was provided five days later at **T5**. Sand flies were killed and preserved in 70% ethanol according to the following five experimental groups:

- **T6INF+** (6 days after <u>infected</u> blood meal, plus a subsequent <u>non-infected</u> second blood meal on day 5);

- **T9INF** (9 days after <u>infected</u> blood meal with no subsequent blood meals);

- **T9INF+** (9 days after <u>infected</u> blood meal, plus a subsequent <u>non-infected</u> second blood meal on day 5);

- **T12INF** (12 days after <u>infected</u> blood meal with no subsequent blood meals);

- **T7CON** sand flies (7 days after <u>non-infected</u> blood meal with no subsequent blood meal) were used as control samples.

## Staining

Fly legs were removed prior to staining to enhance stain penetration. Initial staining of sand flies (n = 18) used a method found to be effective on larger flies, 1% iodine staining [13]. However, we encountered problems with the rapid movement of iodine out of such small vectors into the ethanol preservative in which the flies were scanned, especially apparent in flies held in the scanning queue for more than 12–24 hours. This produced low quality scans, unsuitable for other than the most basic analysis, i.e. to determine if infected or not infected. Iodine-based stain solutions penetrate quickly into the soft tissues, but they can also dilute rapidly and the contrast persists only for short periods [18]. In addition, iodine staining sometimes results in blurry edges of some key tissues and structures [19]. To increase X-ray absorption of low-density tissues and enhance overall contrast, subsequent sand flies were stained for one week by immersion in a solution of 1% phosphotungstic acid (PTA) in 80% ethanol, with continual swirling on a laboratory mixer and a change to fresh staining solution mid-period [16]. Unlike the iodine stain, PTA staining penetrates more slowly into the tissues, but it binds permanently and results in sharp tissue differentiation [18,19], so material could be held for a lengthy period before scanning without loss of stain.

## Micro-CT scanning

Seventeen sand fly specimens, three from each of the five experimental groups plus two uninfected T9+ flies, were stained and then scanned in the Core Research Laboratories at the Natural History Museum, London. For scanning, each stained specimen was placed in 80% ethanol in a 0.2 ml micro centrifuge tube. To minimise movement during scanning, each specimen was held in place at the bottom of its tube by inserting a plug of green floral foam (phenol-formaldehyde foam) over the specimen. Specimens were scanned in a Zeiss Versa 520 system (Carl Zeiss Microscopy GmbH, Jena, Germany) with 4× optical magnification, with exposure set to 12 s, current to 83 µA and voltage to 60 kV. The resulting projections were reconstructed as tiff stacks with a voxel size of 0.9 µm$^3$. Reconstructed data were imported into VG Studio Max 2.2 (Volume Graphics GmbH, Heidelberg, Germany), in which slice stacks were rendered, re-oriented and visualized in the three principal planes (cross, horizontal and sagittal). The reconstructed data were also loaded into Avizo Version 2019.1 (Visualization Sciences Group, Bordeaux, France) for segmentation, linear measurement, volumetric measurement and 3D visualization. All uses of 'segmentation' and 'segmented' in this study refer to the manual or automated process of assigning a label to all pixels in a 2D section, hence all voxels in a 3D model, that correspond to a certain structure. This process enables that structure to be separated from the pixels/voxels of other tissues and of the background [20], so that it can be visualized in 3D and its volume can be measured by, respectively, selecting and counting the labelled voxels.

## Scanning electron microscopy

Sand flies for scanning electron microscopy (SEM) were dissected in 70% ethanol and then pinned to small sheets of plastic and kept in ethanol until study. They were critically point dried in a Balzers CPD 030 and then coated with gold-palladium (20 nm) in a Cressington 208HR sputter coater. They were imaged on a ZEISS Ultraplus Field Emission Scanning

electron microscope (FEGSEM) at up to x 15K in the Core Research Laboratories at the Natural History Museum, London.

### Statistical analysis

The low number of replicates in this pilot study did not enable a rigorous statistical analysis to be performed. Therefore, the results were presented in an exploratory way with descriptive statistics, using Unistat Statistical Package version 10.11 to tabulate or visually compare all raw values of the five groups of flies (i.e. displaying more information than just means and standard errors) and to present the graphical output of a Principal Components Analysis.

## Results

### Level of *Leishmania* infection in sand flies

The mean numbers of *Leishmania mexicana* parasites in the midgut of sand flies in groups T9INF and T9INF+ were recorded to assess the effect of a second blood meal. Flies in group T9INF had a mean of 16,100 parasites per fly (n = 15; SD = 9,938) whereas those in group T9INF+ had a mean of 51,840 parasites per fly (n = 15; SD = 12,824). These means were statistically significantly different (unpaired t-test, p<0.0001). Before and after these time points, flies in the primary infection had a mean of 10,320 parasites per fly in group T6INF (n = 10; SD = 11,240) and 9,081 parasites per fly in group T12INF (n = 10; SD = 9,024). Those means were also both statistically significantly different to the mean of the T9INF+ group (unpaired t-test, P<0.005).

### Visualization of PSG-parasite plug

Our results clearly show the grossly distended midgut of infected sand flies (Fig 1C–1E), with a plug of PSG and parasites, in contrast to the thin and flattened midgut of uninfected individuals with no plug (Fig 1A and 1B). One of the uninfected flies was in the T9+ group, but on scanning it was clearly shown to have no plug and, therefore, was relabelled as T9CON (Fig 1B). In contrast, the PSG-parasite plug was clearly discernible in infected flies, with a distinct edge that could be readily segmented in most cases. In just one infected sand fly the PSG-parasite plug was diffuse in the abdominal midgut and it was difficult to segment in that region. Therefore, our segmentation of the plug was of the most 'compact' part and would have slightly underestimated the total volume as any diffuse elements would not have been included. For clear 3D visualisation of organs, false-colouring was applied during segmentation and its benefits are shown when removed (Fig 1F). No 3D images were constructed without segmentation.

The stomodeal valve of the uninfected flies was always shut or narrowly opened (0–17 μm diameter; Table 1). However, in infected flies the stomodeal valve was always clearly open (15–60 μm diameter; Table 1, Figs 2E and 3D) and obstructed with the plug of PSG and parasites, which frequently also extended into the oesophagus and the pharynx, filling them either partially or completely; some infected sand flies after a second blood meal (T9INF+) or after a lengthy first infection (T12INF) also had the PSG-parasite plug filling the anterior portion of the crop (Table 2 and Fig 2E). The distension of the thoracic midgut and the volume of the PSG-parasite plug were especially great in the infected sand flies that had taken a second blood meal (T9INF+) (Figs 1 and 2). In two of the three scanned T9INF+ infected sand flies, a constriction in the thoracic region of the midgut was observed (Fig 1D).

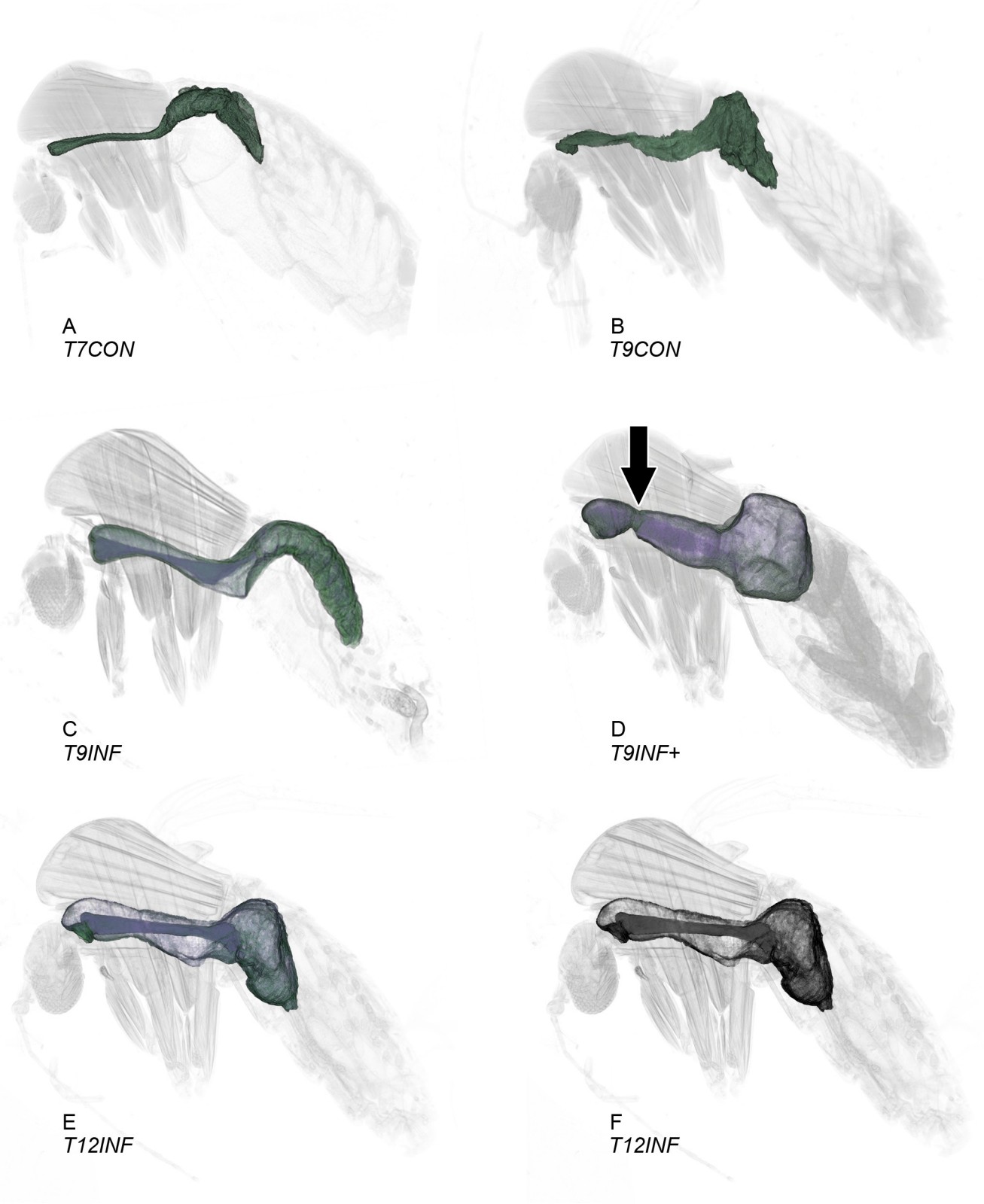

A
*T7CON*

B
*T9CON*

C
*T9INF*

D
*T9INF+*

E
*T12INF*

F
*T12INF*

**Fig 1. False-colour 3D-surface models of *L. longipalpis* females after different feeding and post-infection periods created by software segmentation (see Methods).** Midgut (green) made transparent to show the PSG-parasite plug (purple) in the infected specimens. Arrow indicates constriction observed in the thoracic midgut of a T9INF+ infected sand fly (D). T7CON (A) and T9CON (B) flies were 7 and 9 days, respectively, after a non-infected blood meal. T9INF (C) and T12INF (E/F) flies were 9 and 12 days, respectively, after an infected blood meal. T9INF+ flies (D) were 9 days after an infected blood meal and 4 days after a second, non-infected blood meal. For comparison the false-colour was removed from T12INF (F).

## Volume measurement of thoracic midgut and PSG-parasite plug

The abdominal midgut volumes of all groups were similar except for the T6INF+ group, which had a blood meal the previous day that was still being digested. This resulted in the T6INF+ group having an average abdominal midgut volume that was 6.6 times greater than the other groups (Table 1). Therefore, further comparisons in this study were based only on data from the thoracic midgut. In most scanned sand flies, the plug of PSG and parasites was concentrated in the thoracic region of the midgut–this was most clearly seen when the segmented PSG-parasite plug was virtually dissected away from the midgut (Fig 3). Volumetric measurements were made of the midgut and of the PSG-parasite plug in the thoracic region (Table 1) and results for all flies in each of the five experimental groups were plotted (Fig 4). Due to the low number of specimens there were no statistically significant differences between groups, but the greatest volumes of parasite-PSG plug were found in the T9INF+ infected group, averaging more than twice the volume of any other group (Fig 4B).

A comparison of the average values of volumetric data shows clearly the effect of infection and of a second non-infected blood meal on PSG-parasite plug volume and on the morphology of the thoracic midgut. The control flies (T7CON) had no parasites and, therefore, no PSG whereas those nine days after an infected blood meal (T9INF) had an average thoracic PSG-parasite plug volume of 0.0013 mm$^3$ (on average 15.5% of the thoracic midgut volume) and those similarly infected but with a second blood meal (T9INF+) had an average thoracic

**Table 1. Measurements from micro-CT micrographs of three individual sand flies in each of the five experimental (infected) or control (uninfected) groups showing thoracic and abdominal midgut and PSG-parasite plug volumes and diameters of the stomodeal valve (SV) opening and of the anterior midgut just posterior to the SV (see Fig 3D for example).** T7CON flies were 7 days after a non-infected blood meal. T9INF and T12INF flies were 9 and 12 days, respectively, after an infected blood meal. T6INF+ and T9INF+ flies were 6 and 9 days, respectively after an infected blood meal and 1 and 4 days, respectively, after a second, non-infected blood meal.

| Group | Infected | Sand fly number* | Thoracic midgut (mm$^3$) | Thoracic PSG-parasite plug (mm$^3$) | Abdominal midgut (mm$^3$) | Abdominal PSG-parasite plug (mm$^3$) | SV opening diameter (µm) | Midgut diameter (µm) |
|---|---|---|---|---|---|---|---|---|
| T6INF + | YES | 1 (21) | 0.005224 | 0.001414 | 0.141556 | 0.000383 | 41.10 | 133.00 |
| | | 2 (31) | 0.004847 | 0.002023 | 0.144471 | 0.000075 | 52.80 | 142.00 |
| | | 3 (40) | 0.005468 | 0.000853 | 0.163616 | 0.000797 | 33.30 | 127.00 |
| T7CON | NO | 4 (27) | 0.003431 | 0 | 0.011395 | 0 | 17.01 | 99.97 |
| | | 5 (32) | 0.000880 | 0 | 0.009200 | 0 | 0 | 47.23 |
| | | 6 (28) | 0.003138 | 0 | 0.012642 | 0 | 0 | 66.50 |
| T9INF | YES | 7 (23) | 0.003500 | 0.000507 | 0.012900 | 0.000091 | 35.14 | 134.36 |
| | | 8 (54) | 0.007573 | 0.001149 | 0.011986 | 0.000412 | 60.00 | 129.00 |
| | | 9 (55) | 0.013196 | 0.002219 | 0.050581 | 0.003560 | 17.54 | 161.88 |
| T9INF + | YES | 10 (25) | 0.007800 | 0.005072 | 0.043900 | 0.010061 | 43.55 | 150.98 |
| | | 11 (37) | 0.012249 | 0.002267 | 0.039612 | 0.000008 | 49.23 | 165.01 |
| | | 12 (35) | 0.013126 | 0.003918 | 0.023858 | 0.001334 | 15.43 | 183.78 |
| T12INF | YES | 13 (29) | 0.004680 | 0.001453 | 0.022400 | 0.000709 | 39.28 | 182.89 |
| | | 14 (38) | 0.006921 | 0.000441 | 0.026188 | 0.001015 | 48.70 | 112.00 |
| | | 15 (39) | 0.005443 | 0.000929 | 0.009106 | 0.000396 | 53.70 | 154.00 |

*Numbers in brackets are experimental codes for each sand fly.

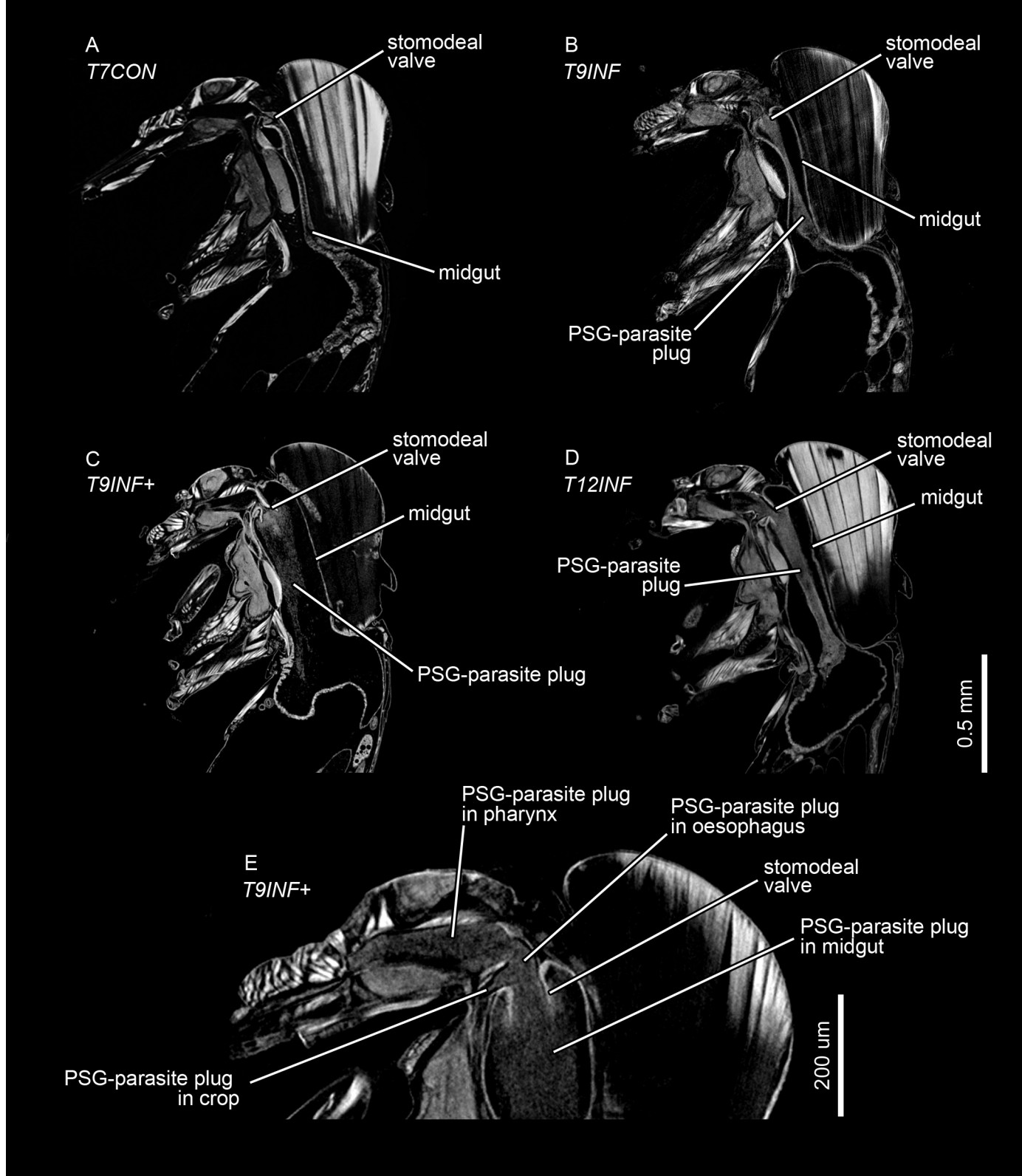

**Fig 2. Micro-CT based virtual sections of *L. longipalpis* females after different feeding and post-infection periods.** T7CON flies were 7 days after a non-infected blood meal. T9INF and T12INF flies were 9 and 12 days, respectively, after an infected blood meal. T6INF+ and T9INF+ flies were 6 and 9 days, respectively after an infected blood meal and 1 and 4 days, respectively, after a second, non-infected blood meal. (A-D) Sagittal sections. (E) Sagittal section of T9INF+ fly, showing PSG-parasite plug in the foregut (oesophagus, pharynx and crop).

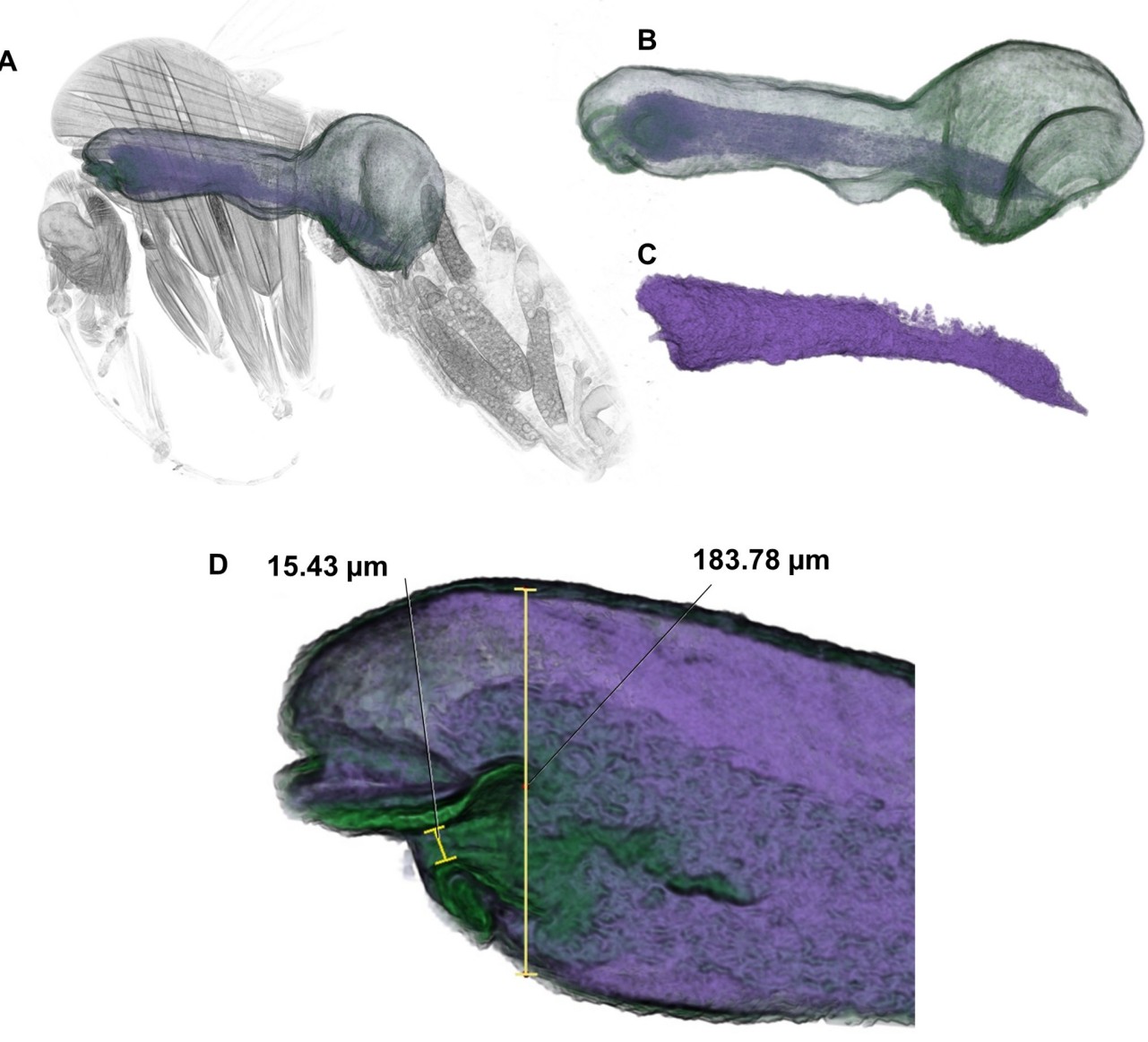

**Fig 3.** False-colour 3D-surface models of the whole fly (A), midgut (B) and PSG-parasite plug (C) of a T9INF+ infected female *L. longipalpis* (9 days after an infected blood meal and 4 days after a second, non-infected blood meal) created by software segmentation (sand fly number 12 (35) in Table 1). Segmentation of the midgut (B, green) out of the fly enabled volume measurement (0.037 mm$^3$; thoracic midgut only = 0.013 mm$^3$). The PSG-parasite plug (C, purple) was then segmented out of the midgut enabling volume measurement (0.0052 mm$^3$, corresponding to 14.2% of the total midgut volume; thoracic midgut only = 0.0039 mm$^3$ and 29.8%). Note the forward projection through the stomodeal valve of the PSG-parasite plug at the anterior (upper) end. The diameter of the stomodeal valve opening and of the midgut immediately posterior to the valve were measured in D.

PSG-parasite plug volume of 0.0038 mm$^3$ (on average 37.8% of the thoracic midgut volume). These average volume measurements confirmed the greater distension of the thoracic midgut and larger volume of the PSG-parasite plug in infected sand flies after a subsequent, second blood meal (T9INF+) compared to specimens that did not have a second blood meal (T9INF), not only to those at the same post-infection period but also to later stage infections (T12INF, average thoracic PSG-parasite plug volume = 0.0009 mm$^3$, 18.1% of the thoracic midgut volume) (Fig 4).

**Table 2. Regions of the sand fly digestive tract anterior to the stomodeal valve (foregut) containing PSG-parasite plug (green, ✓) or no plug (red, ✗).** T7CON flies were 7 days after a non-infected blood meal. T9INF and T12INF flies were 9 and 12 days, respectively, after an infected blood meal. T6INF+ and T9INF+ flies were 6 and 9 days, respectively after an infected blood meal and 1 and 4 days, respectively, after a second, non-infected blood meal.

| Group | Infected | Sandfly Number* | Oesophagus | Pharynx | Crop |
|---|---|---|---|---|---|
| **T6INF+** | YES | 1 (21) | ✓ | ✓ | ✗ |
| | | 2 (31) | ✓ | ✗ | ✗ |
| | | 3 (40) | ✗ | ✗ | ✗ |
| **T7CON** | NO | 4 (27) | ✗ | ✗ | ✗ |
| | | 5 (32) | ✗ | ✗ | ✗ |
| | | 6 (28) | ✗ | ✗ | ✗ |
| **T9INF** | YES | 7 (23) | ✓ | ✗ | ✗ |
| | | 8 (54) | ✓ | ✗ | ✗ |
| | | 9 (55) | ✗ | ✗ | ✗ |
| **T9INF+** | YES | 10 (25) | ✓ | ✓ | ✓ |
| | | 11 (37) | ✓ | ✓ | ✓ |
| | | 12 (35) | ✓ | ✗ | ✗ |
| **T12INF** | YES | 13 (29) | ✓ | ✓ | ✓ |
| | | 14 (38) | ✓ | ✓ | ✗ |
| | | 15 (39) | ✓ | ✓ | ✓ |

*Numbers in brackets are experimental codes for each sand fly.

A similar comparison of linear measurements around the stomodeal valve region of the thoracic midgut (Table 1 and Fig 3D) showed a clear difference between uninfected and infected flies, but a less clear difference between infected flies that either had a second non-infected blood meal or had not. Thus, average stomodeal valve opening diameter (and anterior midgut diameter) values were 5.7 μm (71.2 μm), 37.6 μm (141.7 μm) and 36.7 μm (166.6 μm) for the T7CON, T9INF and T9INF+ groups, respectively.

A principal component analysis of the five groups of sand flies based on four thoracic variables (the volumes of the thoracic midgut and of the thoracic parasite-PSG plug and the diameters of the stomodeal valve opening and of the thoracic midgut) showed that several groups could be clearly separated from one another using just the first two components, which together accounted for 86.1% of the total variance, especially the uninfected control flies (T7CON) and those in the group of infected flies 4 days after their second blood meal (T9INF +) (Fig 5). The infected groups that were 6 days (T6INF), 9 days (T9INF) and 12 days (T12INF) post-infection without a second blood meal completely overlapped and formed a third cluster. The T9INF group included a single outlier fly (number 9) which had values more similar to those of the T9INF+ group (Fig 5 and Table 1).

### Visualization of *Leishmania* within the sandfly

Examination of midgut homogenates by light microscopy confirmed that the infections were present at all days sampled. The SEM study showed large numbers of parasites in the pharynx of T9INF+ sand flies (Fig 6). Based on their morphology, they are predominantly leptomonad promastigotes.

## Discussion

Visualization of 3D models of segmented sand fly midguts supported the hypothesis that the midgut is deformed by a mass of parasites embedded in promastigote secretory gel (PSG)

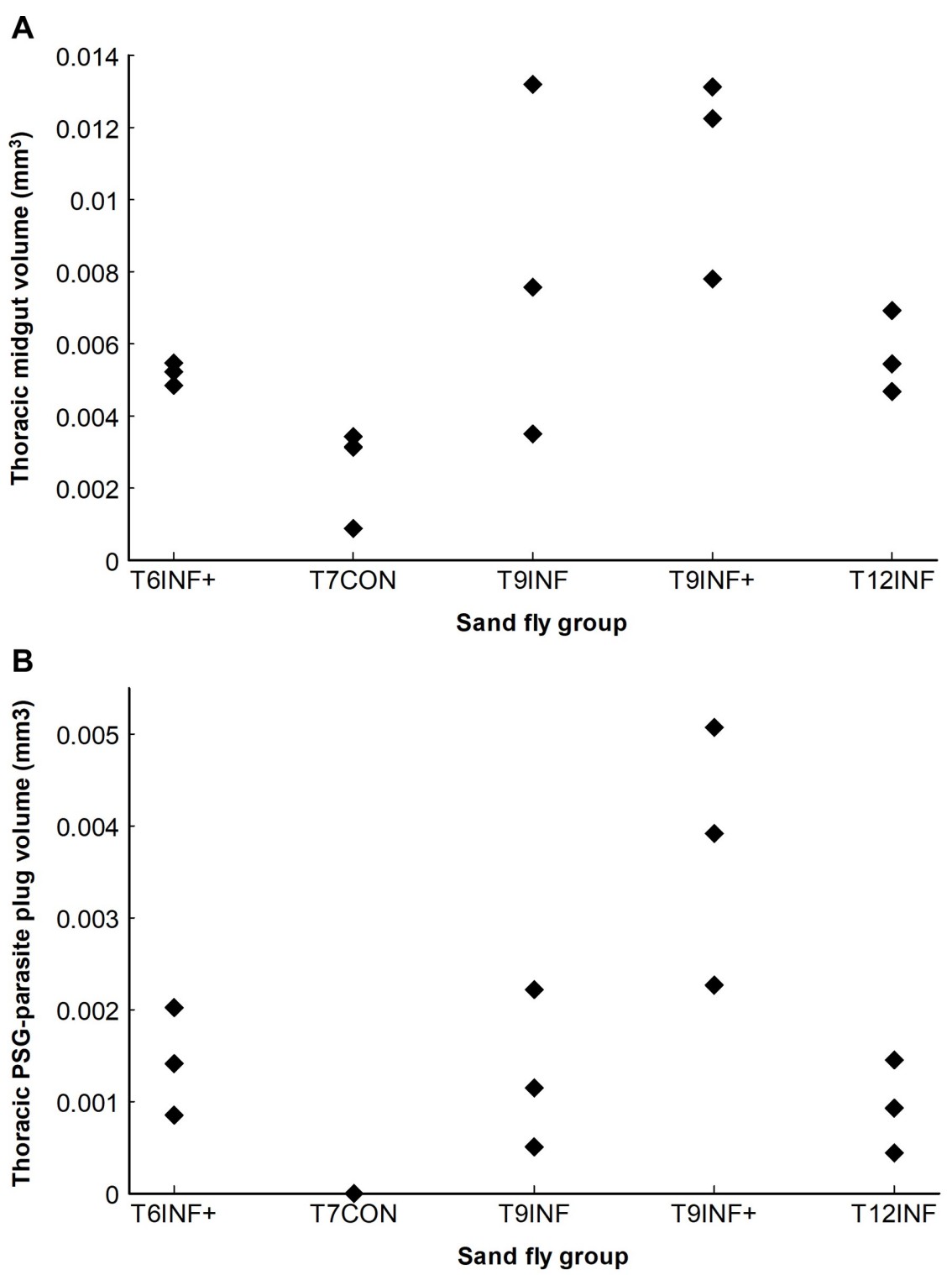

**Fig 4.** Volumes (mm³) of the thoracic midgut (A) and of the thoracic PSG-parasite plug (B) from five groups of female sand flies, *Lutzomia longipalpis* (n = 3 per group), either 6, 7, 9 or 12 days after first feeding on an infected blood meal (INF) or an uninfected blood meal (CON), for two groups with a second, uninfected blood meal five days after the first blood meal (+). Note how this second blood meal for group T9INF+ increased their thoracic midgut and, especially, PSG-parasite plug volumes compared to other groups.

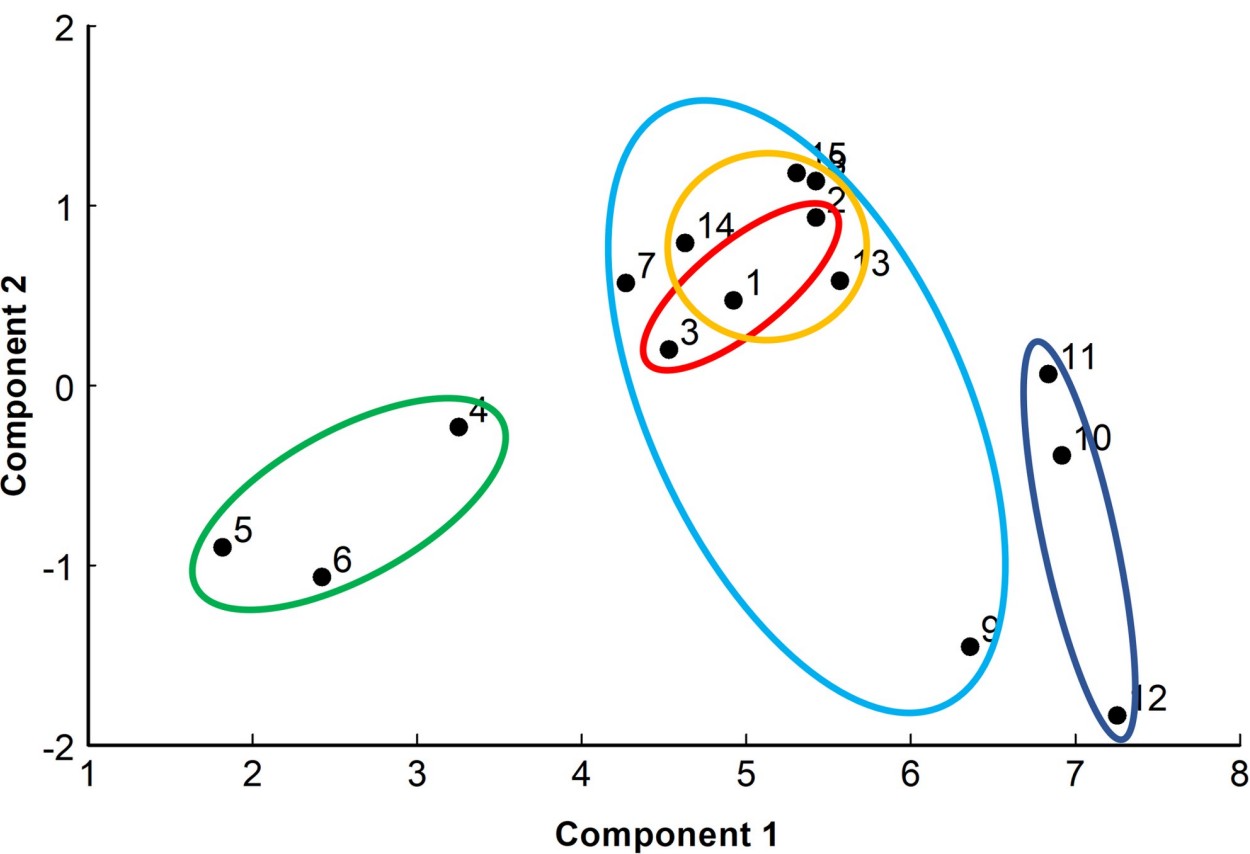

**Fig 5. Plot of principal components 1 (63.4% of variance) and 2 (22.8% of variance) from principal component analysis of five groups of female sand flies, *Lutzomia longipalpis*, either 6, 7, 9 or 12 days after first feeding on an infected blood meal (INF) or an uninfected blood meal (CON), for two groups with a second, uninfected blood meal five days after the first blood meal (+).** Hand-drawn ovals enclose all group points, i.e., T6INF+ (red), T7CON (green), T9INF (pale blue), T9INF+ (dark blue) and T12INF (gold). Variables included were thoracic midgut volume, thoracic PSG-parasite volume and the diameters of the opening of the stomodeal valve and of the midgut just posterior to the stomodeal valve.

secreted by the *Leishmania* parasites. The PSG-parasite plug was clearly observed within the midgut of infected sand flies, opening and blocking the stomodeal valve and distending and enlarging the sand fly thoracic midgut, particularly around the region dorsal to the valve. The PSG-parasite plug was also observed to extend into regions of the foregut, including oesophagus, pharynx and crop, especially of sand flies with a lengthy infestation (T12INF)–a second blood meal facilitated this (T9INF+) (Table 2). Combined with the finding that a second blood meal will generate many more metacyclics [5], this finding argues for more infectious bites following a second blood meal.

The scans of infected flies (Fig 1C–1F), illustrate the difficulty that the bloated, "blocked fly" must have in normal feeding, reflecting the fact that heavily infected flies take an average 2.4 times longer to feed than uninfected flies [21] and are more likely to obtain only a partial blood meal [3]. The PSG-parasite plug and the deformation of the midgut have been imaged here at a much greater resolution than anything hitherto published. Moreover, the volumes of the thoracic midgut and of the PSG-parasite plug of all fly groups were measured, showing higher values in infected sand flies (T6INF+, T9INF, T9INF+ and T12INF) compared to uninfected ones (T7CON). Additionally, comparing these volumes between groups of infected sand flies showed higher values in T9INF+ specimens compared to infected sand flies without a subsequent blood meal, both at the same (T9INF) and later (T12INF) post-infection times

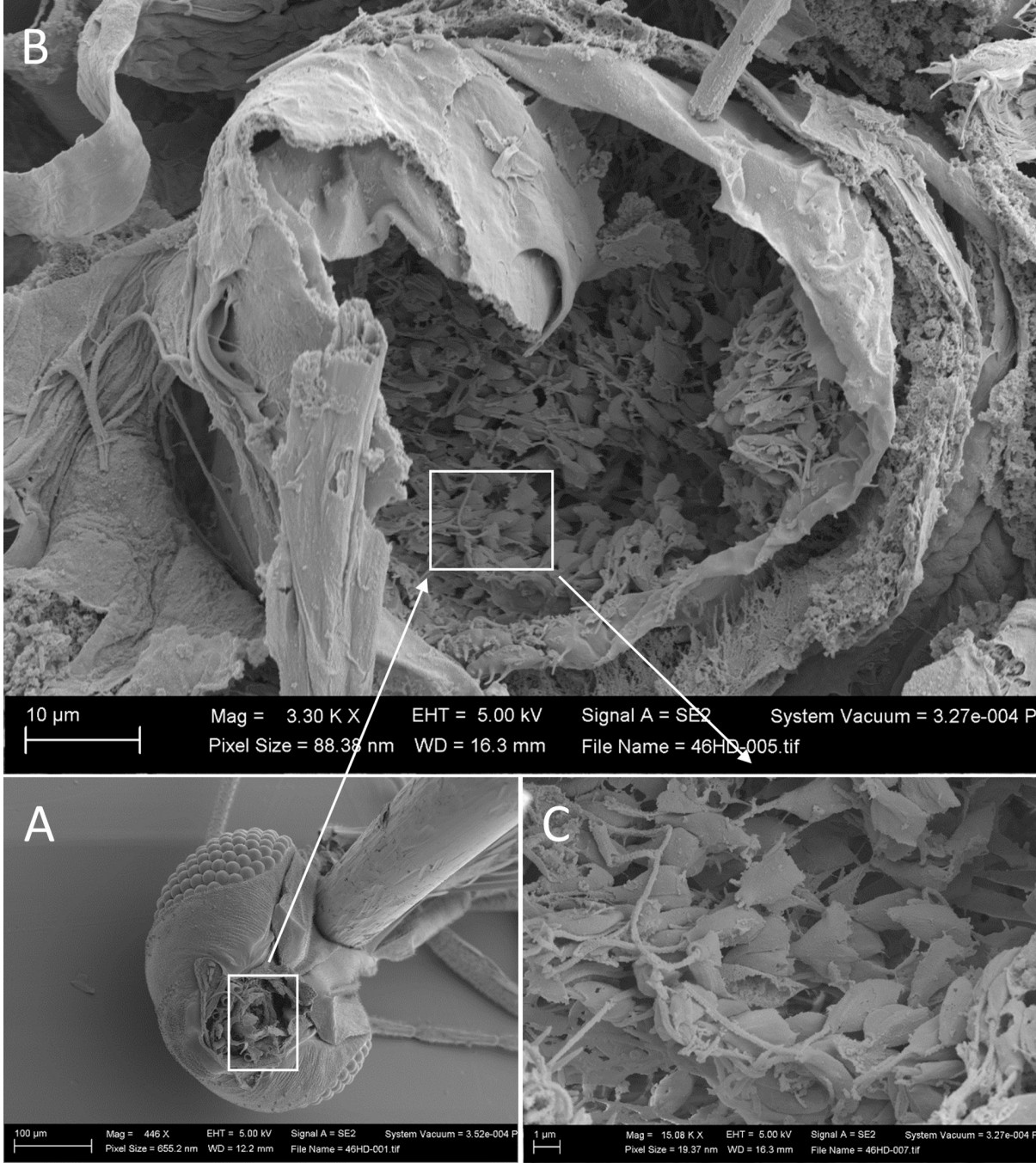

**Fig 6. Scanning electron micrographs of the posterior surface of the dissected head of a sand fly nine days after an infective blood meal and four days after a second uninfected blood meal, T9INF+.** A) pinned head showing orientation on SEM stub enabling view into pharynx; B) view into the dissected pharynx showing large numbers of *Leishmania mexicana* parasites; C) close-up of *L. mexicana* parasites.

(Fig 4). It has been suggested that sand flies that do not take subsequent blood meals produce poor infections [5], which is in accordance with the relatively low PSG-parasite plug volumes observed here in infected T9INF and T12INF sand flies (Table 1 and Fig 4B).

It should be noted that the PSG-parasite plug volumes we reported could have underestimated the actual plug volumes and, therefore, underestimated the percentage of the gut

occupied by PSG and parasites. Firstly, we were unable to measure the volumes of diffuse portions of the PSG-parasite plug or of any discrete, thread-like filaments [22] separated from the main plug. Secondly, it is possible that preservation in ethanol caused shrinkage of the PSG because there appeared to be more of a gap between the PSG-parasite plug and the gut wall than is seen in fresh dissections (MER, personal observation). This possibility needs to be checked, however, if PSG shrinkage in ethanol does occur it is likely to be of a similar degree whatever the volume of PSG, so comparison of the volumes between different fly groups preserved in the same way should still provide a good indicator of the relative volumes of the PSG-parasite plug.

Individual parasites could not be visualized at the resolution used for scanning entire sand flies (pixel size = 0.9–1.0 μm). Even at higher resolution (pixel size = 0.3 μm) parasites were not unambiguously identified, although the different radio-opacity of shapes within the gel suggested the presence of parasites. However, the high-resolution micro-CT scans required a scan time of 58.4 hours–this could not be justified for routine midgut scans as it carried the risk of specimen movement leading to blurring. *Leishmania* parasites associated with the thoracic midgut and stomodeal valve can be imaged by fluorescent and confocal microscopy [23], but only after destructive dissection. For non-destructive imaging of *Leishmania* in intact vectors, synchotron nanotomography [24] could be a potential way forwards, following appropriate specimen preparation, allowing greater resolution than micro-CT (e.g. pixel size of 10 nm [25]) in a shorter scan time. This technique has been used to study insects [26], plants [27] and malaria parasites infecting red blood cells [25]. Nevertheless, in our study *Leishmania* parasites were observed by SEM of the dissected midgut of infected sand flies that had previously been scanned by micro-CT, confirming their infection (Fig 6).

Compared to uninfected sand flies, the greater the degree of opening of the stomodeal valve of infected sand flies and the greater diameter of the thoracic midgut just posterior to the stomodeal valve might be related to the binding of haptomonads to the valve or to pathology of the valve caused by chitinases of the *Leishmania* parasites, resulting in its deformation [5,10,28–30], and to mechanical blockage of the valve by the PSG plug [3]. A second blood meal has also been shown to expand the number of *L. infantum* haptomonads in *Lu. longipalpis* [4,5], which can result in the formation of a haptomonad parasite sphere–a prominent 'ball' of haptomonads attached to the stomodeal valve, resulting in a massive expansion of the valve [5]. Although we did not observe such a structure in the limited number of *L. mexicana*-infected *Lu. longipalpis* subjected to micro-CT, its combination with the expanded PSG-parasite plug would significantly contribute to the regurgitation of parasites during blood feeding.

Both physical blockage and valve pathology seem to favour parasite transmission by causing regurgitation from the vector during blood feeding, but vector infectivity ultimately depends on its parasite burden. In this sense, a subsequent blood meal on day 5 post-infection has been observed to result in an increase in the parasite burden by day 9 post-infection, in comparison to infected sand flies without a subsequent blood meal [5]. Such an increase is proposed to be caused by dedifferentiation of metacyclic promastigotes into a replicative leptomonad-like stage [5]. Although we were limited by the small number of replications, our preliminary results lend support to those observations and suggest that leptomonad promastigotes or leptomonad-like forms, the so-called 'retroleptomonad promastigotes' [5], may also secrete high concentrations of the filamentous proteophosphoglycans that form the scaffold of the PSG in response to a second blood meal (Figs 3, 4B and 6), thus contributing to augment vector infectivity.

Micro-CT and SEM study of second blood-fed flies showed heavy infections in the pharynx. We identified them as leptomonad promastigotes as they appeared to be unattached and to have a broad cell body, 10–12 μm in length with a long flagellum. However, it is possible that a

proportion may have been retroleptomonads, although it is difficult to say with certainty as they present similar morphology to leptomonad promastigotes [5]. Retroleptomonads are proposed to be the product of dedifferentiation of metacyclic promastigotes in response to serum factors in the second blood meal and are largely responsible for reinitiating growth of the infection to provide many more metacyclics when the fly seeks a further blood meal some 5–6 days later. We speculate that after a second blood meal the expansion of the PSG blockage can anchor the multiplying leptomonads and retroleptomonads in the anterior midgut to complete their metacyclogenesis and facilitate their access to the foregut and mouthparts for onward transmission. Accumulation of more parasites and PSG in this region could also interfere with the functioning of sensilla which probably regulate engorgement and result in altered blood-feeding behaviour [31]. To this end, *Lu. longipalpis* with a primary infection were more persistent in blood-feeding, and were more likely to re-feed on a host or a near by host when interrupted when they had mature infections with metacyclics and PSG in the anterior midgut [21]. As a result, this altered biting behaviour encouraged more transmissions when the flies were most infectious. It remains to be tested whether this behavioural manipulation is exaggerated following a second blood meal. Studies such as these with other *Leishmania*-sand fly combinations will prove interesting to elucidate the role of the PSG-parasite plug in vector behaviour and transmission, as the amounts of PSG are likely to vary.

In addition, the observation of a constriction in the thoracic region of the midgut of two of the three T9INF+ flies could be further evidence of the impact of a second non-infected blood meal and deserves closer attention.

In conclusion, our pilot study shows the potential of micro-CT techniques to provide new insights into the study of parasite-vector interactions and provides support to the hypothesis that a second non-infected blood meal enhances parasite transmission. Visualization of individual *Leishmania* parasites was not possible by micro-CT at the resolution used in our study, however, micro-CT can be a powerful tool to record parasite induced changes in the 3D morphology of the vector midgut and sand fly anatomy, both qualitatively and quantitatively, without physical damage to the specimens.

## Acknowledgments

We are grateful to Dr Davide Pigoli (King's College, London, UK) for his advice on the statistical interpretation of our data, to Dr Vincent Fernandez (Natural History Museum, London, UK) for discussions on micro- and nanotomography and to the two journal reviewers for their thoughtful and constructive comments and suggestions.

## Author Contributions

**Conceptualization:** Martin J. R. Hall, Debashis Ghosh, Daniel Martín-Vega, Brett Clark, Robert A. Cheke, Matthew E. Rogers.

**Formal analysis:** Martin J. R. Hall, Debashis Ghosh, Daniel Martín-Vega, Brett Clark, Matthew E. Rogers.

**Funding acquisition:** Martin J. R. Hall, Debashis Ghosh, Daniel Martín-Vega, Robert A. Cheke, Matthew E. Rogers.

**Investigation:** Martin J. R. Hall, Debashis Ghosh, Daniel Martín-Vega, Brett Clark, Innes Clatworthy, Matthew E. Rogers.

**Project administration:** Martin J. R. Hall.

**Visualization:** Martin J. R. Hall, Debashis Ghosh, Daniel Martín-Vega, Brett Clark, Innes Clatworthy.

**Writing – original draft:** Martin J. R. Hall, Daniel Martín-Vega, Matthew E. Rogers.

**Writing – review & editing:** Martin J. R. Hall, Debashis Ghosh, Daniel Martín-Vega, Brett Clark, Innes Clatworthy, Robert A. Cheke, Matthew E. Rogers.

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
