## [Decision Letter · Decision Letter 0]

11 Jun 2021

Dear Dr. Hall,

Thank you very much for submitting your manuscript "Micro-CT visualization of promastigote secretory gel (PSG) in the digestive tract of the sand fly Lutzomyia longipalpis infected with Leishmania mexicana" for consideration at PLOS Neglected Tropical Diseases. As with all papers reviewed by the journal, your manuscript was reviewed by members of the editorial board and by several independent reviewers. Based on the reviews, we are likely to accept this manuscript for publication, providing that you modify the manuscript according to the review recommendations.

We cannot make any decision about publication until we have seen the revised manuscript and your response to the reviewers' comments. 

Sincerely,

Guy Caljon

Associate Editor

Anthony Papenfuss

Deputy Editor

Reviewer's Responses to Questions

Key Review Criteria Required for Acceptance?

Methods

-Are the objectives of the study clearly articulated with a clear testable hypothesis stated?

-Is the study design appropriate to address the stated objectives?

-Is the population clearly described and appropriate for the hypothesis being tested?

-Is the sample size sufficient to ensure adequate power to address the hypothesis being tested?

-Were correct statistical analysis used to support conclusions?

-Are there concerns about ethical or regulatory requirements being met?

Reviewer #1: The authors introduced a new innovative method, the 3D reconstruction using Micro-CT scanning, to visualize infected sand flies.

The Micro-CT scanning was based on a limited sample of 17 sand fly specimens, divided into 5 experimental groups. Therefore, the results are not sufficient for rigorous statistical analyses and the authors correctly used mainly descriptive and exploratory statistics to present and visualize data. Formally, the PCA analysis should be based on higher sample size (at least 30 are recommended). However, the separation of the groups is clear and meaningful (Fig 5). The first two components account for almost 70%, so I agree with the presentation of these data.

Reviewer #2: Methodology is well described and details provided.

Results

-Does the analysis presented match the analysis plan?

-Are the results clearly and completely presented?

-Are the figures (Tables, Images) of sufficient quality for clarity?

Reviewer #1: The morphological differences among individual groups (infected flies vs control flies, infected flies with the second bloodmeal vs infected flies without the second bloodmeal) are clearly visible from the figures and well supported with measurements presented in Table 1. The only point at issue may be the nature of the purple stained segments – is it really just PSG? In my opinion, these areas are promastigotes surrounded with PSG and the method does not allow distinguishing how big part corresponds to PSG vs parasites.

The authors performed the analysis of leishmania numbers in individual groups of infected sand flies (lines 111-114), but the results are not presented.

Reviewer #2: 4. In line 261: The statement “Based on their morphology, they are predominantly leptomonad promastigotes.” Urges for further discussion. Are these parasites the retroleptomonads? What are they doing in the pharynx? 

5. In the results section, there is no description of the volumes of the abdominal midguts. Anyhow, there is no point of comparing the volumes of abdominal midguts as the blood meals bias the comparisons of uninfected vs infected. In addition, it also skewed the PCA analysis. Therefore, PCA analysis should be done and presented without the abdominal midgut data. 

8. In figure 1: Please, provide a model without midgut/psg staining

9. Figure 2 caption needs to include the differences between time points. T12INF is also in the figure. Should describe what “+” stands for.

10. Figure 5: Needs to include the variance of each component either in the caption or in the graph axis’ legends

Conclusions

-Are the conclusions supported by the data presented?

-Are the limitations of analysis clearly described?

-Do the authors discuss how these data can be helpful to advance our understanding of the topic under study?

-Is public health relevance addressed?

Reviewer #1: See the General Comments section

Reviewer #2: Regarding the overall conclusion of the manuscript: What guarantees to the authors that the structures they visualized with micro-CT and named PSG are in fact PSG rather than leishmania parasites? If micro-CT's resolution is not powerful enough to visualize leishmania, how can the authors then postulate that PSG can be visualized in their samples? In Serafim et al, 2018, similar structures encompass massive amounts of parasites in the midgut rather than the light diffraction caused by the presence of massive amounts of fPPG. 

6. In lines 278-279, the statement “the volumes of the midgut” should be replaced by “the volumes of the thoracic midgut”, as that was the region of the midgut significant for the study.

7. In line 309: The statement “mechanical blockage of the valve by the PSG plug [3].” Should be followed by the sentence “or by the binding of haptomonads to the Stomodeal Valve (Volf et al 2004, Serafim et al, 2018).

11. In lines 324-330: Are there other technical possibilities of using microCT to visualize Leishmania? Is any kind of leishmania labeling available for this technique? If so, please discuss.

<b>Editorial and Data Presentation Modifications?

Reviewer #1: Line 41: I suggest adding the taxonomical category (Diptera: Psychodidae) behind Phlebotomine sand flies. 

Line 53: I suggest omitting the word “clearly” as it is redundant and questionable in case of haptomonads

Line 57: I suggest modifying the sentence to: It is the mass of parasites and PSG that obstruct….

Line 185: Here and throughout the text of the manuscript, I suggest to modify the “PSG plug” to just a “plug” (for explanation see General Comments). 

Line 240: stomodeal valve opening diameter

Reviewer #2: (No Response)

Summary and General Comments

Reviewer #1: 

The manuscript is well written and innovative. It is a pilot study bringing a novel non-invasive method to the research of Leishmania-sand fly interactions. Using Micro-CT, authors visualized the 3D picture of the midgut of sand flies infected with Leishmania parasites in situ for the first time. Although the study is based on limited sample size (understandable for technical reasons), it clearly demonstrated that the gut of infected sand flies was distended in comparison with uninfected control flies and the second non-infected bloodmeal contributed to this distension significantly. Highly valuable are the measurements showing that the opening of the stomodeal valve was significantly higher in infected flies.

 I have two objective issues:

• The nature of the plug (purple stained segments): Why authors homologize the plug with PSG only? The substantial part of the plug is formed by Leishmania promastigotes (haptomonads and leptomonads). If authors cannot provide the evidence that the method visualizes the PSG only, then, throughout the manuscript, the plug should be interpret as a formation consisting of both PSG and parasites, 

• Authors should provide the numbers of parasites in the studied vector groups.

Reviewer #2: 

The manuscript # PNTD-D-21-00697, submitted by Hall and colleagues, describe the establishment of the Micro-CT visualization technique to observe morphological structures of sand flies infected with leishmania. The implementation of technological advances for the study of vector morphology and vector-pathogen interaction needs to be complemented, and by themselves such advances are of interest for the vector biology community. Nonetheless, no evidence is provided that the structures the authors claim to be visualized are in fact PSG. Also, the manuscript needs in multiple instances to include important findings by other authors. Altogether, these issues preclude me to provide a more positive recommendation for the manuscript at this stage. Therefore, my recommendation is for Major Revision.

Major Concerns

1. Line 48: The statement “recent research on an older observation” is merely assumptive - Authors have no evidence that such a research was done based on such an older observation. There was other evidence pointing to the same direction. Statement should be modified.

2. Line 57: Regarding the statement “It is the PSG that obstructs the sand fly gut [8,9]” :

 a. Statement is misleading. The role of PSG in blocking the sand fly gut is restricted thus far to L. mexicana in Lutzomyia longipalpis. There is no scientific evidence that enough PSG is produced by other species of Leishmania to block their vectors’ stomodeal valve. Authors should either spell out their leishmania and sand fly species in each sentence where they are mentioned or state that such findings about the PSG are yet to be demonstrated in other leishmania-sand fly pairs, and should replace citation (8, 9) by appropriate literature - Schlein 1992, Volf et al 2004

 b. In Volf et al 2004, it is shown that the attachment of heptomonad-like parasites onto the cuticle liner of the Stomodeal Valve is responsible for the blocking effect, including in Trypanosoma, which has also been noticed in Serafim et al, 2018 and denominated the “Haptomonad Parasite Sphere (HPS)”. Paragraph should be modified to incorporate such findings.

 c. In Schlein 1991,1992, author(s) showed that the damage of the stomodeal valve by leishmania was responsible for the blockage mechanisms. Authors need to add this information as well. 

3. In line 59: In the statement “so-called “blocked fly hypothesis” [3,10].”, the authors mis-cited the studies that postulated the hypothesis, which are Schlein et al. (1991, 1992).

PLOS authors have the option to publish the peer review history of their article (https://journals.plos.org/plosntds/s/editorial-and-peer-review-process#loc-peer-review-history). If published, this will include your full peer review and any attached files.

Do you want your identity to be public for this peer review? For information about this choice, including consent withdrawal, please see our Privacy Policy (https://www.plos.org/privacy-policy).

Reviewer #1: No

Reviewer #2: Yes: Iliano V. Coutinho-Abreu
---

## [Editor Report · Decision Letter 1]

16 Jul 2021

Dear Dr. Hall,

Thank you very much for submitting your manuscript "Micro-CT visualization of a promastigote secretory gel (PSG) and parasite plug in the digestive tract of the sand fly Lutzomyia longipalpis infected with Leishmania mexicana" for consideration at PLOS Neglected Tropical Diseases. As with all papers reviewed by the journal, your manuscript was reviewed by members of the editorial board. Based on the reviews, we are likely to accept this manuscript for publication, providing that you modify the manuscript according to the review recommendations. 

Sincerely,

Guy Caljon

Associate Editor

Anthony Papenfuss

Deputy Editor

One comment of reviewer 2 remains improperly addressed: "The statement "recent research on an older observation" is merely assumptive - Authors have no evidence that such a research was done based on such an older observation. There was other evidence pointing to the same direction. Statement should be modified." Although authors indicate in the answer letter that this statement was removed, this indicated change was not made in the revised manuscript.

Some minor typographical comments:

Line 99: Le. mexicana -> L. mexicana

Lines 245, 253: T12 INF -> T12INF

Figure Files:

Data Requirements:

Reproducibility:

References

---

## [Editor Report · Decision Letter 2]

27 Jul 2021

Dear Dr. Hall,

We are pleased to inform you that your manuscript 'Micro-CT visualization of a promastigote secretory gel (PSG) and parasite plug in the digestive tract of the sand fly Lutzomyia longipalpis  infected with Leishmania mexicana' has been provisionally accepted for publication in PLOS Neglected Tropical Diseases.

Best regards,

Guy Caljon

Associate Editor

Anthony Papenfuss

Deputy Editor

---

## [Editor Report · Acceptance letter]

12 Aug 2021

Dear Dr. Hall,

We are delighted to inform you that your manuscript, " Micro-CT visualization of a promastigote secretory gel (PSG) and parasite plug in the digestive tract of the sand fly Lutzomyia longipalpis  infected with Leishmania mexicana ," has been formally accepted for publication in PLOS Neglected Tropical Diseases.

Best regards,

Shaden Kamhawi

co-Editor-in-Chief

Paul Brindley

co-Editor-in-Chief
